# Identification of the Tembusu Virus in Mosquitoes in Northern Thailand

**DOI:** 10.3390/v15071447

**Published:** 2023-06-27

**Authors:** Rodolphe Hamel, Ronald Enrique Morales Vargas, Dora Murielle Rajonhson, Atsushi Yamanaka, Jiraporn Jaroenpool, Sineewanlaya Wichit, Dorothée Missé, Anamika Kritiyakan, Kittipong Chaisiri, Serge Morand, Julien Pompon

**Affiliations:** 1MIVEGEC, Université de Montpellier, IRD, CNRS, 34394 Montpellier, France; dorothee.misse@ird.fr (D.M.); serge.morand@umontpellier.fr (S.M.); julien.pompon@ird.fr (J.P.); 2Department of Clinical Microbiology and Applied Technology, Faculty of Medical Technology, Mahidol University, Nakhon Pathom 73170, Thailand; sineewanlaya.wic@mahidol.ac.th; 3Viral Vector Joint Unit, Join Laboratory, Mahidol University, Nakhon Pathom 73170, Thailand; 4Department of Medical Entomology, Faculty of Tropical Medicine, Mahidol University, Bangkok 10400, Thailand; ronald.mor@mahidol.ac.th (R.E.M.V.); dorarajonhson@gmail.com (D.M.R.); 5Department of Pharmacology, Faculty of Science, Mahidol University, Bangkok 10400, Thailand; 6Research Institute for Microbial Diseases, Osaka University, Osaka 565-0871, Japan; knmya@biken.osaka-u.ac.jp; 7Department of Medical Technology, School of Allied Health Sciences, Walailak University, Nakhon Si Thammarat 80160, Thailand; jjirapor@mail.wu.ac.th; 8Excellent Center for Dengue and Community Public Health, Walailak University, Nakhon Si Thammarat 80160, Thailand; 9Faculty of Veterinary Technology, Kasetsart University, Bangkok 10900, Thailand; anamikanose.k@gmail.com; 10Department of Helminthology, Faculty of Tropical Medicine, Mahidol University, Bangkok 10900, Thailand; kittipong.cha@mahidol.ac.th

**Keywords:** mosquito-borne viruses, Tembusu virus, Culex mosquito, emergent arboviruses

## Abstract

Among emerging zoonotic pathogens, mosquito-borne viruses (MBVs) circulate between vertebrate animals and mosquitoes and represent a serious threat to humans via spillover from enzootic cycles to the human community. Active surveillance of MBVs in their vectors is therefore essential to better understand and prevent spillover and emergence, especially at the human–animal interface. In this study, we assessed the presence of MBVs using molecular and phylogenetic methods in mosquitoes collected along an ecological gradient ranging from rural urbanized areas to highland forest areas in northern Thailand. We have detected the presence of insect specific flaviviruses in our samples, and the presence of the emerging zoonotic Tembusu virus (TMUV). Reported for the first time in 1955 in Malaysia, TMUV remained for a long time in the shadow of other flaviviruses such as dengue virus or the Japanese encephalitis virus. In this study, we identified two new TMUV strains belonging to cluster 3, which seems to be endemic in rural areas of Thailand and highlighted the genetic specificities of this Thai cluster. Our results show the active circulation of this emerging flavivirus in Thailand and the need for continuous investigation on this poorly known but threatening virus in Asia.

## 1. Introduction

Mosquito-borne viruses (MBVs) of zoonotic origins are responsible for multiple animal and human diseases worldwide and represent a large reservoir of viruses with emergence potential via spillover from their enzootic cycles. Tembusu virus (TMUV) is an emerging mosquito-borne flavivirus that belongs to the Ntaya serocomplex, including Ntaya virus, Bagaza virus and Israel Turkey virus (refer to the ICTV database—https://ictv.global/report/chapter/flaviviridae/flaviviridae/orthoflavivirus (accessed on 26 June 2023)). Similar to other flaviviruses, including DENV and JEV, TMUV is an enveloped, positive-sense single-stranded RNA virus with an approximately 11-kb genome (reviewed in [1]). Despite recent sporadic outbreaks, knowledge on TMUV ecology and biology remains fragmented, precluding a thorough evaluation of its emergence potential.

TMUV was first isolated in Malaysia in 1955 [2], before being reported in different surveys in Asia and Southeast Asia (SEA) including China, Malaysia, Taiwan and Thailand [3,4,5,6]. TMUV has been intermittently reported in wild and domestic birds and in trapped mosquitoes [7,8]. TMUV infects a wide variety of avian species such as ducks, geese, chickens, sparrows and pigeons [1]. Since 2000, new variants of TMUV have been reported to cause several outbreaks in poultry and birds. Symptoms include dramatic decreases in egg production, severe neurologic disorders and retarded growth [5,7]. Migration of wild birds close to poultry farms could allow transmission to domestic ducks, while retention of the virus in high-density duck-producing areas could facilitate the rapid spread of the disease. Because of the symptom severity in ducks and the economic importance of ducks, some reports named the new viral variant Duck-TMUV (DTMUV) [3]. Nonetheless, hereafter, we will use TMUV as a generic term to refer to all viruses belonging to the TMUV phylogenetic group. TMUVs are phylogenetically divided into two lineages: the “TMUV lineage” including the original viruses, and the “DTMUV lineage”. The DTMUV lineage is divided into three different clusters, named “TMUV cluster-1”, “TMUV cluster-2” (with sub-cluster a and b) and “TMUV cluster-3” [9]. *Culex* mosquitoes are likely the main vector of transmission, as TMUVs have been isolated from several *Culex* species such as *Culex tritaeniorhynchus*, *Cx. Vishnui*, and *Cx. Gellidus* [1,10]. In addition to vector transmission, vertical transmission and non-vector transmission in birds (by air droplet exposure or by close contact) are suspected [11,12,13].

Located in the heart of South East Asia, Thailand has tight interactions with surrounding countries, including China and Laos. Endemic transmission of numerous mosquito-borne flaviviruses such as JEV and DENV occurs in Thailand, and other arboviruses associated with diseases in humans [14,15]. Thailand is largely covered with forests and rural areas, with an increasing entanglement of rural and urban territories and a densification of urban areas. In Thailand, TMUV was isolated in mosquitoes from the rural parts of the country, including the provinces of Kamphaengphet [8], Chiang Mai [16] and Kanchanaburi [5,17]. TMUV strains were also detected in broiler and layer ducks from the provinces of Chonburi, Nakhom Pathom, Nakhon Ratchasima, Prachinburi and Signburi [5]. Such recurrent detection indicates a wide distribution of TMUV in Thailand. Accordingly, in 2013, TMUV outbreaks occurred throughout the year (August 2013–September 2014) and many farms were affected, leading to losses in the poultry industry. Alarmingly, seroconversion was detected in humans, irrespective of contact with ducks, suggesting a zoonotic emergence of TMUV [18]. However, the serological survey in humans was conducted with a limited number of samples, and the survey lacked methodological details. In this context, TMUV surveillance in animals and vectors is essential to prevent agro-economical losses and evaluate emergence in humans.

In this study, we conducted agnostic arbovirus surveillance in mosquitoes along an ecological gradient in the Thai northern province of Nan, which shares a border with Lao PDR. The sub-district of Saenthong in the province of Nan is divided into two geographical landscape types: an agricultural lowland, including low-density urbanized villages, and a highland with sparse villages and an agricultural zone embedded in the forest zone located close to the protected Nanthaburi National Park. This contrasted area is separated by a transition zone including agricultural areas with rice paddy fields and dwellings. These landscapes provided an ideal study area with low and high levels of human-impacted habitats to study the ecology of MBVs and their mosquito vectors. In the different landscapes we sampled, we detected TMUV and insect-specific flaviviruses in the transition area, and in the lowland as well as the forest. We further characterized the phylogeny of the new TMUV strains, revealing a potential endemic cluster in Thailand.

## 2. Materials and Methods

### 2.1. Mosquito Collections

Mosquitoes were collected in the province of Nan in the northern part of Thailand (Figure 1a). The survey area was located in the Saen Thong sub-district of the Tha Wang Pha district, a rural area localized in an ecological gradient between forests, paddy fields lowlands and peridomestic urban areas. The eight collection points were distributed along a transect covering eight villages and a forested area (three sessions) (Figure 1a,b). The research proposal, involving specimen collection in Nan province, was approved by the Faculty of Tropical Medicine, Mahidol University, under agreement number FTM ECF-033-00.

Samples were collected using BG-sentinel traps combined with BG Lure (Biogents AG, Regensburg, Germany). BG-sentinel traps operated for 48 h, for day- and night-time sessions.

Mosquito specimens were transported in cold boxes containing frozen cold packs to the field laboratory, for sorting up to the species identification level. In cases in which species could not be determined, specimens were grouped according to their genus only, and noted “*Genus*” sp. Mosquitoes were sorted and pooled in a 15 mL tube according to genus, sex and collection site, and then stored frozen. The samples were then transported in a liquid nitrogen tank to the department of Medical Entomology, Faculty of Tropical Medicine, Mahidol University, where samples were identified up to the species level following the morphological identification keys outlined by Rattanarithikul, R. et al. [19,20,21]. Mosquitoes were identified on a chilled table set to −4 °C, then mosquitoes were pooled according to species, sex and collection site and stored in a −80 °C freezer.

Mosquito pools (1 to 15 specimens per pool) were made according to mosquito species, sex and collection location. Stainless steel beads (5 mm diameter) were added to tubes containing mosquitoes before homogenizing using a TissueLyzer (Qiagen, Hilden, Germany) at 50 cycles/s for 5 min in 500 µL of DMEM medium (Gibco, Waltham, MA, USA), complemented with 1% of penicillin/streptomycin solution (Gibco, Waltham, MA, USA) and 1× of Fungizone solution (Gibco, Waltham, MA, USA). After homogenization, an additional 1 mL of DMEM medium was added. Tubes were clarified using centrifugation at 13,000× *g* at 4 °C for 10 min. The supernatants were collected and filtered, using an 0.2 µm syringe filter (Sartorius, Bangkok, Thailand), into 1.5 mL tube and stored at −80 °C before RNA extraction.

### 2.2. RNA Extraction and Reverse Transcription

Viral RNA was extracted from the supernatant of mosquito homogenate using a NucleoSpin^®^ virus kit (Macherey-Nagel, Düren, Germany) according to the manufacturer’s protocol. Briefly, a 200 µL of homogenized sample was lysed in 5 µL of proteinase K and 200 µL lysis buffer containing guanidine hydrochloride. Carrier RNA was then added to the mixture, and the viral nucleic acid was then extracted and collected in an elution volume of 30 µL of RNase-free water. Purified RNA extracts were stored at −80 °C until virus screening using RT-PCR. Reverse transcription (RT) was performed using an M-MLV reverse transcriptase kit (Promega, Madison, WI, USA) on 14 µL of an RNA sample, following the manufacturer’s instructions. cDNA was stored at −20 °C until subsequent analyses.

### 2.3. Detection of Flaviviruses and Alphaviruses Using PCR

The pan-flavivirus primers [22] PFlav-fAAR (5′-TACAACATGATGGGAAAGAGAGAGAARAA-3′) and PFlav-rKR (5′-GTGTCCCAKCCRGCTGTGTCATC-3′) were used to amplify a 256 base pair(bp) region of the NS5 gene of Flaviviruses. PCR was performed using GoTaq G2 Master Mix (Promega, Charbonnières-les-Bains, France) and 2 µL of cDNA with the following parameters: 95 °C for 3 min, 45 cycles of 95 °C 15 s, 56 °C 15 s, 72 °C 20 s and 72 °C for 2 min. PCR products were visualized on 1.8% agarose gel. Amplicons were purified from gel using a PureLink Gel extraction kit (Thermo Fisher Scientific, Illkirch-Graffenstaden, France) and stored at −20 °C.

The pan-alphavirus primers [23] PanAlpha F2A forward primer (5′-ATGATGAARTCIGGIATGTTYYT-3′), and reverse primers R2A (5′-ATYTTIACTTCCATGTTCATCCA-3′), R3A (5′-ATYTTIACTTCCATRTTCARCCA-3′), R4A (5′-ATYTTIACTTCCATGTTGACCCA-3′) were used to amplify a 200-pb region of the nsP4 gene in the alphavirus genome. PCR was performed using GoTaq G2 Master Mix (Promega, Charbonnières-les-Bains, France) and 2 µL of cDNA with the following parameters: 95 °C for 3 min, 45 cycles of 95 °C 15 s, 54 °C 15 s, 72 °C 20 s and 72 °C for 2 min. PCR products were visualized on 1.8% agarose gel.

All purified amplicons obtained with pan-flavivirus- or pan-alphavirus-PCR were characterized using Sanger sequencing in both forward and reverse directions (Eurofins, Vergèze, France). Sequence identities were determined via BLAST alignment (https://blast.ncbi.nlm.nih.gov/Blast.cgi, accessed on 26 June 2023).

### 2.4. TMUV Envelope Sequencing

A 1503-bp amplicon covering the entire TMUV envelope gene was amplified using the primers TMUV-E_F (5′-TTCAGCTGTCTGGGGATGCA-3′) and TMUV-E_R (5′-GGCATTGACATTTACTGCCA-3′). PCR amplification was conducted from 2 µL of cDNA using Q5 High Fidelity DNA Polymerase (NEB, Évry-Courcouronnes, France) and the following parameters: 98 °C for 1 min, 40 cycles of 98 °C 10 s, 60 °C 15 s, 72 °C 60 s and 72 °C for 2 min. PCR products were visualized on 1.5% agarose gel, and amplicons were gel-purified using a PureLink Gel extraction kit (Thermo Fisher Scientific, Illkirch-Graffenstaden, France) and stored at −20 °C. Purified amplicons were sequenced via Sanger sequencing in both forward and reverse directions (Eurofins, Vergèze, France).

### 2.5. Phylogenic Analysis

To characterize TMUV isolated from mosquito homogenates, sequences encoding TMUV Envelope were subjected to phylogenetic analysis, along with representative TMUV sequences obtained from the NCBI GenBank database (Table 1).

Reference sequences were selected to cover all the diversity of TMUV strains and a broad range of geographical origins. All sequences were referenced into the phylogenetic tree in a format consisting of “accession number_country_year of isolation”. Multiple sequence alignments and edits were carried out using MEGA 11. Sequences were edited and sites that could not be unambiguously aligned were excluded from the analyses. Maximum likelihood trees were constructed using PhyML software [24,25] with the best-fit nucleotide substitution model (GTR+G) identified by Akaike’s information criterion (AIC). The bootstrap method was used to estimate the robustness of nodes with 1000 iterations. Phylogenetic trees were edited using FigTree v1.4.4 software. Sequences of Zika virus (ZIKV) (GenBank number: KY766069), JEV (GenBank number: NC001437) and West Nile virus (WNV) (GenBank number: NC009942) were used as the outgroup to provide a relative framework for analyzing the phylogenetic differences between viruses within the Ntaya complex. All sequences from this study have been deposited in the GenBank database, and their accession numbers are shown in Table 1 Amino acid sequences were aligned using the MEGA 11 program to identify specific amino acid variations in the envelop protein sequences of TMUV strains.

## 3. Results

### 3.1. Collection of Mosquitoes

Mosquitoes were collected in eight villages and in one forested area (over three sessions) from the 19th to 26th of July 2019 in the Saenthong sub-district of the province of Nan, Thailand (Figure 1b). The Saenthong sub-district is located in the rural district of Thawangpha. Samples were collected along an ecological gradient from lowland areas, including villages, farms, and agricultural areas (village 1, 2, 3 and 8), to highland areas wherein three villages (village 5, 6 and 7) are located, with small agricultural zones surrounded by a vast forest area. Village 4 is located in a transition zone between lowland and highland (Figure 1b).

A total of 596 mosquitoes were collected and homogenized in 116 pools (Table 2). genera total of 5 genera of *Culicidae* were reported, including 12 species. Some specimens were damaged during collection, making it impossible to identify the species. In these cases, specimens have been listed as “*Genus*” sp. (Table 2). Overall, the *Culex* genus was the most represented (75%, n = 436), with *Cx. vishnui* representing the most prevalent species (39.8%, n = 237). The *Aedes* genus represented 15.4% (n = 92), and *Armigeres* 8.9% (n = 53), while *Mansonia* and *Toxorhynchites* both represented 0.17% (n = 1). However, the mosquito genera largely varied depending on location. All *Toxorhynchites* (n = 1), 78.3% of *Aedes* (n = 72) and 50.9% of *Armigeres* (n = 27) mosquitoes were caught in the forested area, whereas only 1.1% of *Culex* (n = 5) were collected in this area. In contrast, *Culex* mosquitoes represented 74% (n = 431) of all mosquitoes collected in the villages. Furthermore, the majority of *Culex* mosquitoes (89%, n = 387) were collected in the lowland villages and in the transition zone to the highland (villages 1–4 and 8, Figure 1b).

### 3.2. Detection of Flaviviruses and Alphaviruses

A total of 116 pools of mosquito homogenates were tested for flaviviruses and alphaviruses (Table 2). Six pools were positive for flaviviruses, and none for alphaviruses. Positive PCR products were sequenced and sequences were identified using the NCBI BLAST^®^ website. Out of six samples, one *Aedes* and one *Culex* genus pool contained sequences related to the *Yunnan Culex flavivirus* (YNCxFV), with a maximum identity of 80.68% and 81.31%, respectively (Table 3). The sequences of one pool of *Aedes aegypti* and one pool of *Culex vishnui* contained the sequence of *Phlebotomus-associated flavivirus* (*PAFV)* with a maximum identity of 98.07% and 98.78%, respectively (Table 3). The pools P#73 with *Cx. vishnui* and P#49 with *Cx* sp. were both collected in “Ban Huak” village 4, and contained a TMUV-like sequence, which we noted as P73_TH_2019 and P49_TH_2019, respectively. The identity score for the first two hits for P49_TH_2019 sequence was 97.36% (coverage = 99%) and 96.04% (coverage = 99%) to TMUV KAN2016 (GenBank access number: KX184310) and TMUV HNU-NX2-2019 (GenBank access number: OP186478), respectively (Table 3 and Appendix A). The P73_TH_2019 sequence was similar, at 98.83% (coverage = 100%) with TMUV_ GX2021 (GenBank access number: OM240641) and 98.44% (coverage = 100%), with TMUV_ SD2021 (GenBank access number: OM240640) (Table 3 and Appendix A). The first ten hits of each sample are visualized in the Appendix A. The alignment of the P73_TH_2019 with P49_TH_2019 shows a very high similarity (96%), suggesting close phylogenic history between the two TMUVs collected in two different mosquito-pools.

### 3.3. Phylogenetic Analysis of the TMUV Isolates

To phylogenetically characterize the two virus isolates from the pools P49_TH_2019 and P73_TH_2019, we analyzed the envelope sequences. Alignment with representatives of TMUVs’ phylogenetic diversity (Table 1) illustrated the distribution of TMUV strains in five distinctives clusters, and their relation to the Ntaya virus (Figure 2a,b) [1].

A first group, named “TMUV”, includes the original strain isolated in 1955 in Malaysia and the MN747003 strain isolated in Taiwan in 2019. The other viruses are divided into four different clusters, named “cluster 1”, comprising strains isolated in Malaysia and Thailand, “cluster 2.a”, including strains isolated in Thailand and China, “cluster 2.b”, corresponding to strains isolated only in China, and “cluster 3”, including strains isolated in Thailand and China. P49_TH_2019 and P73_TH_2019 isolates from this study form a monophyletic lineage closely related to strains belonging to the cluster 3. The strains in cluster 3 were isolated in Thailand in 2016 and in China in 2014. Furthermore, we generated another phylogenetic tree by including the partial sequences of the envelope gene of three other TMUV strains isolated in Thailand in 1992 and 2002 [10,16]. All three Thai TMUV strains were clustered into the “cluster 3” with the isolates from this study (Figure 2b).

### 3.4. Identification of Envelope Amino Acid Modifications Specific to the TMUV Isolates from Nan Province

The genomic sequences of the P49_TH_2019 and P73_TH_2019 samples cover the entire coding sequence of the envelope protein (E protein), and were used to reveal amino acid differences from other strains belonging to every cluster of TMUV. The positions of amino acids with characteristic substitutions are shown in Table 4. All TMUV clusters have specific amino acid variations at certain positions, with unique patterns for every cluster (Table 4). The strains of cluster 1 carry specific amino acids on position 52 and 83, except for the strain KX097989, isolated in Malaysia in 2012, which has unique substitutions at position 373, 390 and 394. The strains of cluster 2 present unique amino acids on position 89, 180,185, 312, 332 and 451. 

The strains of cluster 3 have nine amino acids unique to this cluster, except for MH748542_CH2014 (Chinese isolate from 2014), and five other amino acids in common with the ancestral cluster “TMUV”. Positions 69, 91, 135, 149, 150, 365, 371, 391 and 394 have common amino acids for the strains MK276427-TH-2016 and P49_TH_2019, and P73_TH_2019 isolates. These amino acids are positioned in the DI (position 135; 149 and 150), DII (position 69) and DIII (position 365; 371; 391 and 394) domains. All TMUV strains present an Asn residue at the position 154; in addition, the strains MK276427_TH_2016, P49_TH_2019 and P73_TH_2019 present an S150N substitution. Finally, both P49_TH_2019 and P73_TH_2019 isolates have two unique substitutions at position 358 (V358I) of the domain III of the envelope protein.

## 4. Discussion

In this study, we report the investigation of more than 596 mosquitoes collected in 2019 in the Nan province of northern Thailand. Samples were collected along a transect covering an ecological gradient ranging from a sparsely urbanized rural area to dwellings clustered in a few villages surrounded by forests and highland. Although we screened for flaviviruses and alphaviruses, we only detected three different flaviviruses in six different mosquito pools. These flaviviruses included two insect-specific flaviviruses and two TMUV strains in two pools of *Culex* mosquitoes.

Composition of mosquito species varied along the ecological gradient depending on the type of landscape. *Aedes albopictus* was abundantly collected in the forest, although a few specimens were captured inside the villages. Our observations were as expected based on its reported peridomestic distribution [26], and were in accordance with previous *Ae. albopictus* collections in the rural and forested habitats of Thailand [27,28]. Additionally, *Ae. albopictus* was the most prevalent species in the forest sample site, although we could not identify all *Aedes* sp. due to sample damage. *Aedes aegypti* was strictly found in village 8, which corresponds to an urbanized rural area in the lowland. *Aedes aegypti* prefers urban zones in part because of its use of human-made containers as breeding sites [29]. *Culex* spp. were the most prevalent genera in the villages, likely in relation to the breeding conditions made available by agricultural activities. We identified *Cx vishnui* in all habitats (from forested to urban areas), as previously reported [30]. In addition to variations in landscape, environmental conditions including altitude can influence mosquito species’ composition and abundance [31]. Accordingly, we reported that 90% of *Culex* spp. were collected in the lowland villages and at the intersection of lowland and highland. Our study provides important information about mosquito species’ distribution in different ecological settings, which will be important when conducting spatial risk analyses for arbovirus circulation.

We were able to identify three different flaviviruses, including two insect-specific flaviviruses: the *Phlebotomus-associated flavivirus* (PAFV) and the *Yunnan Culex flavivirus* (YNCxFV). Interestingly, both viruses were detected in Culex and Aedes spp. These viruses were previously identified in other regions in *Cx. gellidus*, *Cx. tritaeniorhynchus*, *Cx. vishnui* and *Cx. quinquefasciatus* [32,33]. There is a growing interest in insect flaviviruses, as two of these have been shown to increase transmission of pathogenic flaviviruses [34], although the mechanism remains elusive. The wide distribution of insect-specific flaviviruses and their potential role in pathogenic virus transmission warrants further studies [35,36].

Importantly, we identified two isolates as belonging to the TMUV group. The isolate P73_TH_2019 was isolated from a pool of *Cx. Vishnui,* and the isolate P49_TH_2019 was isolated from *Culex* spp. in the same village located in the transition zone between the highlands and lowlands. Although the species of the pool of *Culex* spp. could not be identified at the species level, it is likely *Cx. Vishnui,* since this species was overwhelmingly present among the other identified mosquitoes at the same site. Previous studies looked at the ability of mosquito vectors to transmit TMUV, and observed that mosquitoes of the genus *Culex* were very competent [10]. Accordingly, in Malaysia, China, Thailand, and Taiwan, TMUV was detected in *Cx. tritaeniorhynchus*, *Cx. vishnui*, *Cx. quinquefasciatus*, *Cx. annulus* and *Cx. pipiens* [1]. Although the transmission capacity of *Culex* mosquitoes seems variable, and a source of discussion [17,37], *Cx. tritaeniorhynchus* was proposed as the principal vector [38]. In Thailand, TMUV infection in *Cx. tritaeniorhynchus* collected in paddy fields was reported in the vicinity of Kamphaeng Phet province both in 1982 and 2002 [8,10]. *Culex vishnui* and *Cx. tritaeniorhynchus* belong to the same subgroup, and are also considered major vectors of JEV.

After its first identification in 1955 in Malaysia, TMUV has only been reported in four different countries in Asia. In Thailand, TMUV has been detected mostly in association with large duck farms in the center of the country [9,39]. In both China and Thailand and in a wetland habitat for waterbirds in a suburban area of Taipei city in Taiwan, the strains isolated were related to cluster 2. In contrast, strains belonging to the cluster TMUV and cluster 1 were found in rural or forest areas in Malaysia and Taiwan [6]. In our study, we identified TMUV strains that belong to cluster 3 in a rural area in northern Thailand. The collection site was in a village of 360 inhabitants, surrounded by rice fields and forested areas downstream to a dam. Prior to the large outbreaks of the 2010s, associated with TMUV cluster 2, in China and Thailand, TMUV cluster 3 had already been identified in Thailand, in rural provinces in the west of the country [2,16]. Although Ninvilai et al. previously reported a cluster 3 TMUV being present on a large duck farm in central Thailand [9], it would seem that viruses affiliated with clusters TMUV, 1 and 3 are found preferentially in rural and forested areas. These locations, including wetlands or rice fields, are particularly favorable to the development of mosquitoes of the genus *Culex*, which are described as major vectors of TMUV, and are also areas with domestic and wild birdlife. Thus, these areas could be favorable for the maintenance and spread of TMUV. Further investigation should be carried out to elucidate the possible relationships between the type of viral strain and the kind of ecological area.

It is interesting to note that, similar to the strain P73_TH_2019 in our study, the strains isolated from mosquitoes in 1982 and 2002 in Thailand also belong to cluster 3. Although TMUV has been detected in large parts of the Thai territory, cluster 3 strains were mostly found in rural areas in the north-west and north parts of the country. Our phylogenetic analysis showed that the two new TMUV strains form a monophyletic group closely related to TMUV cluster 3. While they do display the specificities of recent TMUV strains, such as the presence of the S156P substitution in «loop 150» region of the envelope protein [40], Thai cluster 3 strains are phylogenetically closer to the TMUV cluster than to clusters 1 and 2. Previous studies have shown the important effect of amino acid substitution sequences in the viral envelope on the virulence and pathogenicity of TMUV [40,41,42]. Recently, Nivilai et al. showed the presence of unique residues in the envelope protein for strains belonging to cluster 3 [9]. The cluster 3 strains identified in our study possess unique amino acid substitutions that are partially different from those described by Ninvilai et al. We found the same residues in position 149, 150, 391 and 394, as previously described by Ninvilai et al., but we also found other amino acid substitutions in positions 69, 91, 135, 365 and 371. As with most flaviviruses, we found an Asn residue at position 154 for the two TMUV isolates identified in our study. The position of this Asn residue forms an N-x-(S/T) glycosylation motif that plays an important role in pathogenicity of flaviviruses [43,44]. In the cluster 3 strains, except for the strain MH748542_2014_Ch, we also found the presence of an Asn residue in position 150, resulting from an S150N substitution. However, this substitution does not lead to the formation of an N-x-(S/T) motif, thus suggesting the absence of glycosylation in this region. Moreover, a unique V358I substitution appears in both strains P49_TH_2019 and P73_TH_2019, whereas the substitution was not present in the cluster 3 strains MK276427_TH_2016 and MH748542_2014_Ch, or in strains isolated from other clusters. Differences in the envelope sequence may stem from selection in either the host or mosquito. Finally, the whole envelope sequence of the strains isolated in 1992 and 2002 is not available, and therefore we are unable to evaluate if the amino acid signature of the cluster 3 strains isolated in Thailand is recent or not.

We would like to propose that the cluster 3 strains circulate at a regional scale between Thailand and the south of China. The local maintenance of these strains could be explained by the presence of the virus in more isolated areas of Thailand, such as the province of Nan, which carry out less trading than the central regions of the country; these central regions are more densely populated and carry out more economic exchanges with other Asian countries. However, the low number of strains from cluster 3 does not provide sufficient hindsight on the evolution of TMUV in Thailand, and further studies should be conducted to evaluate the impact of the different TMUV clusters on its dissemination, the evolution of pathogenicity and TMUV emergence risk in humans.

In conclusion, we report the detection of TMUV in *Culex* mosquito populations in northern Thailand. Our phylogenetic analysis classified these isolates into TMUV cluster 3, and highlighted the genetic specificities of this cluster, providing insights into the diversity and evolution of TMUV. TMUV surveillance is particularly important in a context of global changes and the intensification of trade between China, Laos, and Thailand. The opening of new economic corridors and new trade routes in the Indo-Pacific region will have a major impact on the emergence of pathogens that were previously restricted to small geographical areas. It is therefore essential to maintain active surveillance for arbovirus emergence and spillover in areas in which there is strong interaction between wildlife, domestic animals, and human communities.

## Figures and Tables

**Figure 1 viruses-15-01447-f001:**
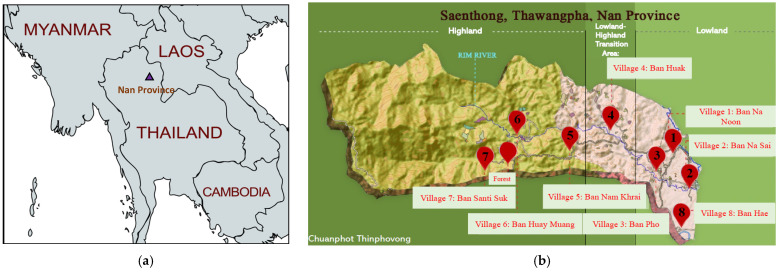
Geographic localization of the study area. (**a**) Localization of Nan province in Thailand. (map created with mapchart.net) and (**b**) localization of the collection sites in the Saen Thong sub-district, Tha Wang Pha district (design credit to Chuanphot Thinphovong) on a schematic flat map reflecting the different areas.

**Figure 2 viruses-15-01447-f002:**
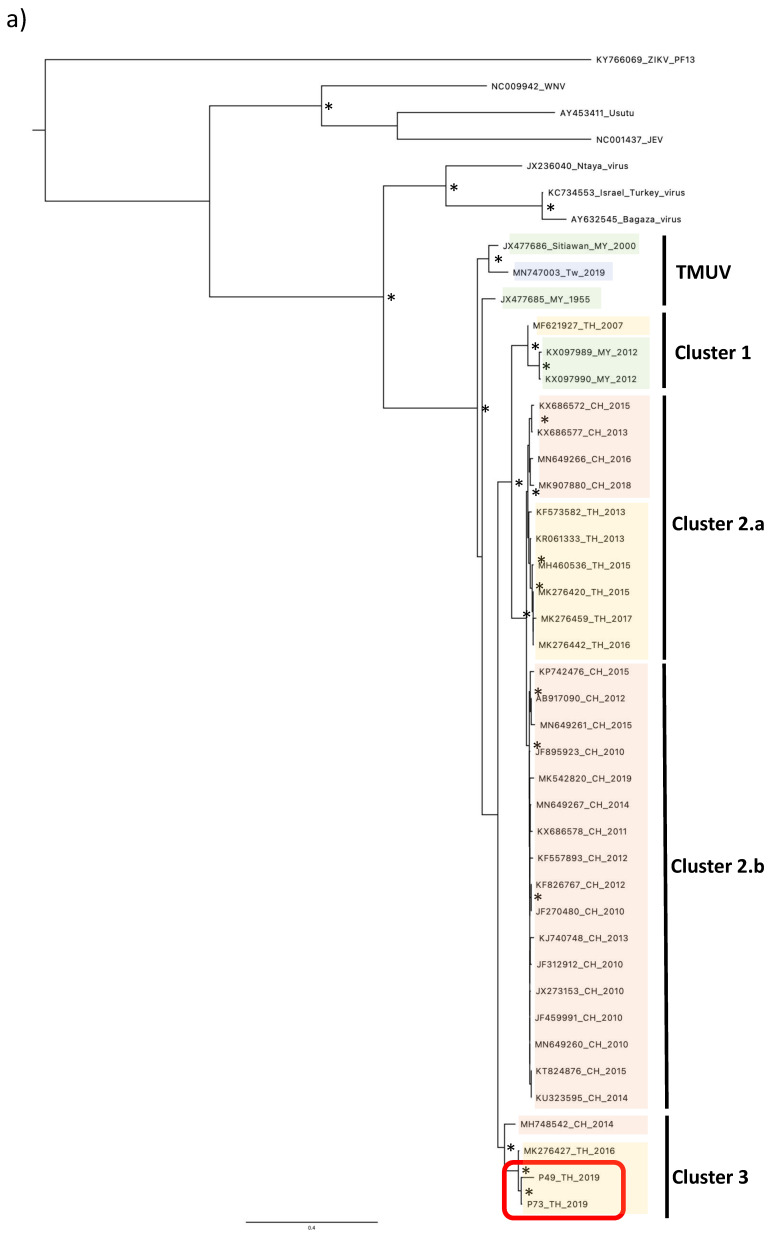
Phylogenetic tree of TMUV isolates from Nan province. Maximum likelihood tree of TMUV envelope sequences, generated using the GTR+G substitution model. Bootstrap values higher than 0.85 are shown on branch nodes by an asterisk. Samples collected in this study are indicated by a red-colored rectangle. (**a**) Phylogenetic tree based on the full envelope gene sequence’s alignment. (**b**) Phylogenetic tree based on the partial envelope gene sequence’s alignment. Country acronyms are abbreviated as CH for China, MY for Malaysia, TW for Taiwan, and TH for Thailand. Strains from this study are marked in red.

**Table 1 viruses-15-01447-t001:** TMUV sequences used in the phylogenetic tree, including the two strains identified in this study.

Virus	GenBank Accession n°	Year	Country
Tembusu virus strains	JX477685	1955	Malaysia
	AB110495	1992	Thailand
	JX477686	2000	Malaysia
	KC810847	2002	Thailand
	KC810846	2002	Thailand
	MF621927	2007	Thailand
	JX273153	2010	China
	JF270480	2010	China
	MN649260	2010	China
	JF895923	2010	China
	JF312912	2010	China
	JF459991	2010	China
	KX686578	2011	China
	KF557893	2012	China
	KF826767	2012	China
	KX097989	2012	Malaysia
	AB917090	2012	China
	KX097990	2012	Malaysia
	KR061333	2013	Thailand
	KJ740748	2013	China
	KF573582	2013	Thailand
	KX686577	2013	China
	MH748542	2014	China
	MN649267	2014	China
	KU323595	2014	China
	KP742476	2015	China
	KX686572	2015	China
	KT824876	2015	China
	MN649261	2015	China
	MK276420	2015	Thailand
	MH460536	2015	Thailand
	MK276427	2016	Thailand
	MK276442	2016	Thailand
	MN649266	2016	China
	MK276459	2017	Thailand
	MK907880	2018	China
	MK542820	2019	China
	MN747003	2019	Taiwan
Ntaya virus	JX236040	2013	-
JEV	NC001437	1989	Japan
WNV	NC009942	1999	USA
ZIKV	KY766069	2013	French Polynesia
Usutu virus	AY453411	2001	Austria
Israel Turkey virus	KC734553	2010	Israel
Bagaza virus	AY632545	2010	Central African Republic
P49_TH_2019 *	ON254216	2019	Thailand
P73_TH_2019 *	OQ543571	2019	Thailand

Year: year of isolation; Country: country of isolation. *: sequence obtained in this study.

**Table 2 viruses-15-01447-t002:** Summary of mosquito species, number of mosquitoes, and pools tested in the study. Mosquitoes for which we only identified the *Genus* were recorded in the column indicated as “*Genus* sp.”.

Collection Site	*Ae. albopictus*		*Ae. aegypti*		*Aedes* sp.		*Cx. quinquefasciatus*		*Cx brevipalpis*		*Cx. hutchinsoni*		*Cx. nigropunctatus*		*Culex* sp.		*Cx. vishnui*		*Cx. tritaeniorhynchus*		*Armigeres* sp.		*Arm.* *kesseli*		*Arm. subalbatus*		*Anopheles* sp.		*An. subpictus*		*Mansonia* sp.		*Toxorhynchites* sp.		Total
	♂	♀	♂	♀	♂	♀	♂	♀	♂	♀	♂	♀	♂	♀	♂	♀	♂	♀	♂	♀	♂	♀	♂	♀	♂	♀	♂	♀	♂	♀	♂	♀	♂	♀	
Village 1	0	0	0	0	0	0	2	6	0	0	0	0	0	0	0	10	0	17	0	0	0	0	0	1	0	0	0	0	0	0	0	0	0	0	36
Village 2	0	2	0	0	0	0	1	5	0	1	0	1	0	0	1	8	1	8	0	0	0	1	0	2	0	0	0	0	0	1	0	0	0	0	32
Village 3	0	2	0	0	0	1	0	0	0	0	0	0	0	0	0	10	1	5	0	0	0	0	0	0	0	0	0	1	0	0	0	0	0	0	20
Village 4	0	0	0	0	0	0	0	1	0	3	0	0	0	0	0	**96** **★**	1	**181** **★**	0	7	0	7	0	5	0	0	0	3	0	3	0	0	0	0	307
Village 5	0	2	0	0	0	0	0	0	0	0	0	1	0	0	0	19	0	19	0	0	0	1	0	4	0	1	0	0	0	0	0	0	0	0	47
Village 6	0	3	0	0	0	1	0	0	0	0	0	0	0	0	1	4	0	0	0	0	0	0	0	0	0	0	0	0	0	0	0	0	0	0	9
Village 7	0	1	0	0	0	2	0	0	0	0	0	0	0	0	0		0	0	0	0	0	1	0	0	0	0	0	0	0	0	0	0	0	0	4
Village 8	1	3	0	**2** **★**	0	0	0	1	0	0	0	0	0	1	2	14	1	2	0	0	0	3	0	0	0	0	0	4	0	2	0	1	0	0	37
Forest session 1	0	15	0	0	0	3	0	0	1	0	0	0	0	0	0	1	0	1	0	0	0	1	0	6	0	1	0	0	0	0	0	0	0	0	29
Forest session 2	0	18	0	0	0	1	0	0	0	0	0	0	0	0	0	**2** **★**	0	0	0	0	0	5	0	4	0	0	0	0	0	0	0	0	0	0	30
Forest session 3	0	8	0	0	0	**27** **★**	0	0	0	0	0	0	0	0	0	0	0	0	0	0	0	6	0	4	0	0	0	0	0	0	0	0	0	1	46
Individuals	1	54	0	2	0	35	3	13	0	4	0	2	0	1	4	164	4	233	0	7	0	25	0	26	0	2	0	8	0	6	0	1	0	1	596
positive pools/Nbr of pool for each species	0/1	0/11	0	**1**/1	0	**1**/8	0/2	0/6	0/1	0/2	0	0/2	0	0/1	0/3	**2**/24	0/4	**2**/24	0	0/1	0	0/8	0	0/7	0	0/2	0	0/3	0	0/3	0	0/1	0	0/1	6/116

★ mosquito collection presenting positive pools for flavivirus family detection.

**Table 3 viruses-15-01447-t003:** Detection and identification of viruses in the mosquito pools.

Mosquito Pool ID	Mosquito Species Identification	Number of Mosquitoes per Pool	First Hit with BLAST^®^ Alignment			
			Collection Site	Viral Identification	Coverage Score	Identity Score
P#13	*Ades aegypti*	2 ♀	Village 8	PAFV	97%	98.07%
P#20	*Aedes* sp.	9 ♀	Forest	YNCxFV	97%	80.68%
P#49	*Culex* sp.	10 ♀	Village 4	TMUV	99%	97.36%
P#60	*Culex* sp.	1 ♀	Forest	YNCxFV	98%	81.31%
P#73	*Culex vishnui*	15 ♀	Village 4	TMUV	100%	98.83%
P#77	*Culex vishnui*	15 ♀	Village 4	PAFV	98%	98.78%

PAFV: Phlebotomus-associated flavivirus; YNCxFV: Yunnan Culex flavivirus; TMUV: Tembusu virus, ♀: female.

**Table 4 viruses-15-01447-t004:** Cluster-specific amino acid substitution for the envelope among TMUV strains. The different domains of the envelope are indicated as DI to DIII.

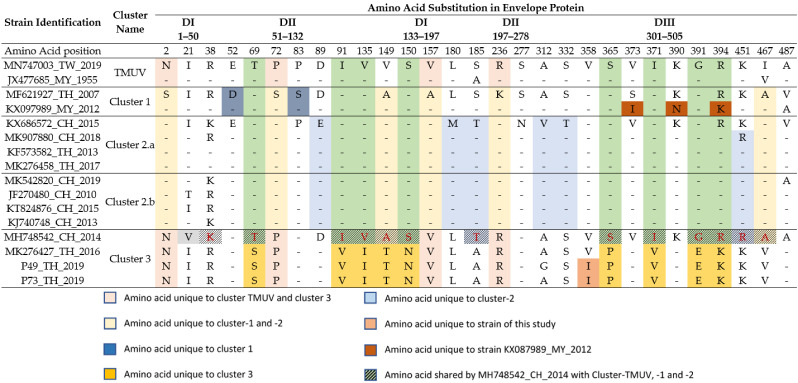

Strain identification: GenBank accession n°#_Country_Year of isolation. The cluster of each strain was determined using the phylogenetic tree in this study. DI: domain I of the envelope protein; DII: domain II of the envelope protein; DIII: domain III of the envelope protein.

## Data Availability

Not applicable.

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
