# Peer review of "Identification of the Tembusu Virus in Mosquitoes in Northern Thailand"

_viruses, 2023, doi:10.3390/v15071447_

Round 1
Reviewer 1 Report
The study is well presented, methods are robust, and the paper is overall well written.
Author Response
20th June 2023
Viruses Editorial Office
Special Issue "Viral Zoonoses: Interactions and Factors Driving Virus Transmission"
Dear Reviewers,
We are very thankful for your time and effort in commenting and improving our manuscript.
Please find below a point-by-point response to each of your comments in blue.
We have addressed each of your comments.
Best regards
Rodolphe Hamel, PhD
Corresponding author
Institute of Research for Development (IRD)
France
Reviewer’s comments
In the manuscript “Identification of Tembusu virus in mosquitoes in northern Thailand” Hamel and colleagues describe and the isolation of two contemporary and novel strains of TMUV in the Nan province of northern Thailand. The authors used field mosquito capture and ecological designation of capture sites across a gradient of urban to sylvan. Diagnostic pan-Alphavirus and pan-Flavivirus primers were used to determine whether a given pool was positive for viruses of the respective genera. The two novel strains underwent phylogenetic characterization based on the coding region of the envelope gene and were compared to sequences from other TMUV strains dating back to the original isolation in 1955. TMUV, particularly DTMUV is a pathogen with enormous emergence potential and studies characterizing its ecology and evolution are well warranted. While the present study does provide some of this, there are several concerns, (including copy-editing for grammar, syntax and clarity) that require addressing prior to publication.
Major Points
- Line 50: TMUV is a member of the Ntaya serocomplex, not “related to”.
We changed “related to” to “belongs”.
- Line 79-80: While the authors correctly point out that a single report has indicated relatively high rates of seroreactivity to endemic TMUV in populations with a) no direct contact with ducks and b) no known association ducks, the study referenced was conducted with limited samples and no description of methodology and inclusion/exclusion criteria. Further, human infection with Ntaya and JEV complex viruses do NOT yield sufficiently high viremia to infect naïve mosquitoes limiting the potential for true zoonotic emergence into a human population. As such, I would suggest the language in this fragment be carefully considered and potentially revised.
As suggested, we now highlight the limitations of the serological survey.
- Materials and Methods, Subsection 2.1: Mosquito Collections: Is Permitting required for mosquito capture; if so, Authors need to note that appropriate permits and licenses were in place and give the permit/license number?
We thank the reviewer for asking this information. In subsection 2.1, we now provide the approval details from Mahidol University to collect specimen.
- la Line 150-151: Please clarify here; as the amplicon size is not described. Later in the paper the authors note the phylogeny was based on the entire coding region for envelope. Please make sure this is captured in methods.
We now detail in subsection 2.4 that the amplicon is 1503bp.
- Lines 169-171 and Figure 2: What is the rationale for including ZIKV, JEV and WNV as outgroups when the Ntaya serocomplex viruses are already included?
We now mention that we included the viruses as outgroup to provide a relative measure in analyzing phylogenetic differences between Ntaya and TMUV.
- Table 2: Please clarify the "Genera sp" columns. In the discussion, authors note that genera level speciation is used due to sample damage (specifically in Aedes). This should be prefaced in the materials and methods and results. Further, if that is only the case in Aedes, why are there columns for Culex sp, Armigeres sp, Anopheles sp, Mansonia sp, and Toxorhynchites sp? The text and figures need to be in agreement.
As suggested, we have clarified the figure legend and the text in the result section.
- Table 2: The total + pools column is 6/116. Based on my count there are 7 positive pools out of 114 total pools (1 pool Aedes albopictus males, 1 pool Aedes aegypti females, 1 pool of Aedes sp females, 2 pools of Culex sp females, and 2 pools of Cx. Vishnui).
We thank the reviewer for pointing to these discrepancies. However, the mistake in the table and we have now corrected that it is “0/1” instead of “1” for Ae albopictus in first column of Table 2. The total number of positive pool is 6 then and we did not change the text.
Further, for the total number of analyzed pools, we forgot to add 1 pool of Cx. brevipalpis and 1 pool for Culex sp in table 2. The total of analyzed pools is 116, as mentioned in the text.
- Fig 2: The inset showing “Isolates from” is useless. The authors have noted the tree shows strains as “accession number_country_year of isolation” in materials and methods. Country acronyms can be noted in the legend.
We have removed the inset and detailed the acronym for the countries.
- Table 4: Below the table, was there supposed to be a color key here? Because it doesn't show up in the PDF.
We are sorry and we do not know why the color did not appear. We suspect an incompatibility between apple and PC as the editorial office can see the color in the table. We have re-inserted the table as a new PDF image and hope the colors are present in this version. We also provide a separate file in .doc for Table 4 to the editorial office.
- Line 248: The results section are to report unfiltered results; the phrase “It is interesting to note…” is best left to the discussion.
We have removed this part of the sentence.
Minor Points (Non-exhaustive, please conduct thorough grammar, syntax, and language editing).
We have thoroughly revised the grammar and text.
- Line 54 and throughout: “emerging potential” is better written as “emergence potential”
We corrected as mentioned.
- Line 59: The word “poultry” by itself can be used as a plural. The statement could better read as “…cause several outbreaks in both poultry and wild birds”.
Corrected.
- Line 69: “were suspected” needs to be corrected to read as “are suspected”
Corrected.
- Line 71: please revise to read as “Endemic transmission of numerous mosquito-borne flaviviruses such as JEV and DENV occurs in Thailand”. Additionally, if other viruses are to be referenced, please note them by name.
Corrected.
- Line 109: “sort it” should be revised to read as “sorting”
Corrected.
- Line 111: “transported in Nitrogen liquid tank” should be revised to read as “transported in a liquid nitrogen tank”
Corrected.
- Line 112: “where was identified” should read as “where samples were identified”
Corrected.
- Line 114: “chill table set up -4°C,” should read as “a chill table set to -4°C,”
Corrected.
- Line 114: “specie” is misspelled. The correct spelling is “species”
Corrected.
- Line 114: “stored at -80°C freeze” should read as “stored in a -80°C freezer”
Corrected.
- Line 134 and 141: First, abbreviations need to be spelled out entirely during their first use. Moreover, “pb” is incorrect, the proper abbreviation is “bp” for base pair. Which in of itself is incorrect because the virus is ssRNA.
Corrected.
- Line 153: I assume a 15% agarose gel is a typo?
Corrected.
- Line 178: “in July 2019” is redundant as the date range of “from the 19th to the 26th of July 2019” is referenced earlier in the same sentence.
Corrected.
- Line 181: “The village 4” should read as “Village 4”
Corrected.
- Line 286: the word “Culex” needs to be italicized.
Corrected.
- Line 295: The phrase “In contrary” should read as “In contrast”
Corrected.
Reviewer 2 Report
In the manuscript “Identification of Tembusu virus in mosquitoes in northern 2 Thailand” Hamel and colleagues describe and the isolation of two contemporary and novel strains of TMUV in the Nan province of northern Thailand. The authors used field mosquito capture and ecological designation of capture sites across a gradient of urban to sylvan. Diagnostic pan-Alphavirus and pan-Flavivirus primers were used to determine whether a given pool was positive for viruses of the respective genera. The two novel strains underwent phylogenetic characterization based on the coding region of the envelope gene and were compared to sequences from other TMUV strains dating back to the original isolation in 1955. TMUV, particularly DTMUV is a pathogen with enormous emergence potential and studies characterizing it’s ecology and evolution are well warranted. While the present study does provide some of this, there are several concerns, (including copy-editing for grammar, syntax and clarity) that require addressing prior to publication.
Major Points
1. Line 50: TMUV is a member of the Ntaya serocomplex, not “related to”.
2. Line 79-80: While the authors correctly point out that a single report has indicated relatively high rates of seroreactivity to endemic TMUV in populations with a) no direct contact with ducks and b) no known association ducks, the study referenced was conducted with limited samples and no description of methodology and inclusion/exclusion criteria. Further, human infection with Ntaya and JEV complex viruses do NOT yield sufficiently high viremia to infect naïve mosquitoes limiting the potential for true zoonotic emergence into a human population. As such, I would suggest the language in this fragment be carefully considered and potentially revised.
3. Materials and Methods, Subsection 2.1: Mosquito Collections: Is Permitting required for mosquito capture; if so, Authors need to note that appropriate permits and licenses were in place and give the permit/license number?
4. Line 150-151: Please clarify here; as the amplicon size is not described. Later in the paper the authors note the phylogeny was based on the entire coding region for envelope. Please make sure this is captured in methods.
5. Lines 169-171 and Figure 2: What is the rationale for including ZIKV, JEV and WNV as outgroups when the Ntaya serocomplex viruses are already included?
6. Table 2: Please clarify the "Genera sp" columns. In the discussion, authors note that genera level speciation is used due to sample damage (specifically in Aedes). This should be prefaced in the materials and methods and results. Further, if that is only the case in Aedes, why are there columns for Culex sp, Armigeres sp, Anopheles sp, Mansonia sp, and Toxorhynchites sp? The text and figures need to be in agreement.
7. Table 2: The total + pools column is 6/116. Based on my count there are 7 positive pools out of 114 total pools (1 pool Aedes albopictus males, 1 pool Aedes aegypti females, 1 pool of Aedes sp females, 2 pools of Culex sp females, and 2 pools of Cx. Vishnui).
8. Fig 2: The inset showing “Isolates from” is useless. The authors have noted the tree shows strains as “accession number_country_year of isolation” in materials and methods. Country acronyms can be noted in the legend.
9. Table 4: Below the table, was there supposed to be a color key here? Because it doesn't show up in the PDF.
10. Line 248: The results section are to report unfiltered results; the phrase “It is interesting to note…” is best left to the discussion.
Minor Points (Non-exhaustive, please conduct thorough grammar, syntax, and language editing)
1. Line 54 and throughout: “emerging potential” is better written as “emergence potential”
2. Line 59: The word “poultry” by itself can be used as a plural. The statement could better read as “…cause several outbreaks in both poultry and wild birds”.
3. Line 69: “were suspected” needs to be corrected to read as “are suspected”
4. Line 71: please revise to read as “Endemic transmission of numerous mosquito-borne flaviviruses such as JEV and DENV occurs in Thailand”. Additionally, if other viruses are to be referenced, please note them by name.
5. Line 109: “sort it” should be revised to read as “sorting”
6. Line 111: “transported in Nitrogen liquid tank” should be revised to read as “transported in a liquid nitrogen tank”
7. Line 112: “where was identified” should read as “where samples were identified”
8. Line 114: “chill table set up -4°C,” should read as “a chill table set to -4°C,”
9. Line 114: “specie” is misspelled. The correct spelling is “species”
10. Line 114: “stored at -80°C freeze” should read as “stored in a -80°C freezer”
11. Line 134 and 141: First, abbreviations need to be spelled out entirely during their first use. Moreover, “pb” is incorrect, the proper abbreviation is “bp” for base pair. Which in of itself is incorrect because the virus is ssRNA.
12. Line 153: I assume a 15% agarose gel is a typo?
13. Line 178: “in July 2019” is redundant as the date range of “from the 19th to the 26th of July 2019” is referenced earlier in the same sentence.
14. Line 181: “The village 4” should read as “Village 4”
15. Line 286: the word “Culex” needs to be italicized.
16. Line 295: The phrase “In contrary” should read as “In contrast”
Minor Points (Non-exhaustive, please conduct thorough grammar, syntax, and language editing)
1. Line 54 and throughout: “emerging potential” is better written as “emergence potential”
2. Line 59: The word “poultry” by itself can be used as a plural. The statement could better read as “…cause several outbreaks in both poultry and wild birds”.
3. Line 69: “were suspected” needs to be corrected to read as “are suspected”
4. Line 71: please revise to read as “Endemic transmission of numerous mosquito-borne flaviviruses such as JEV and DENV occurs in Thailand”. Additionally, if other viruses are to be referenced, please note them by name.
5. Line 109: “sort it” should be revised to read as “sorting”
6. Line 111: “transported in Nitrogen liquid tank” should be revised to read as “transported in a liquid nitrogen tank”
7. Line 112: “where was identified” should read as “where samples were identified”
8. Line 114: “chill table set up -4°C,” should read as “a chill table set to -4°C,”
9. Line 114: “specie” is misspelled. The correct spelling is “species”
10. Line 114: “stored at -80°C freeze” should read as “stored in a -80°C freezer”
11. Line 134 and 141: First, abbreviations need to be spelled out entirely during their first use. Moreover, “pb” is incorrect, the proper abbreviation is “bp” for base pair. Which in of itself is incorrect because the virus is ssRNA.
12. Line 153: I assume a 15% agarose gel is a typo?
13. Line 178: “in July 2019” is redundant as the date range of “from the 19th to the 26th of July 2019” is referenced earlier in the same sentence.
14. Line 181: “The village 4” should read as “Village 4”
15. Line 286: the word “Culex” needs to be italicized.
16. Line 295: The phrase “In contrary” should read as “In contrast”
Author Response

(The authors gave the same response as above.)
